# Therapeutic Atmosphere in Psychotherapy Sessions

**DOI:** 10.3390/ijerph17114105

**Published:** 2020-06-09

**Authors:** Marte L. Siegel, Eva M. Gullestad Binder, Hanne Sofie J. Dahl, Nikolai O. Czajkowski, Kenneth L. Critchfield, Per A. Høglend, Randi Ulberg

**Affiliations:** 1Department of Psychology, University of Oslo, Forskningsveien 3A, 0373 Oslo, Norway; h.s.j.dahl@psykologi.uio.no (H.S.J.D.); n.o.czajkowski@psykologi.uio.no (N.O.C.); 2Vestfold Hospital Trust, Halfdan Wilhelmsens alle 17, 3103 Tønsberg, Norway; 3Department of Graduate Psychology, James Madison University, Harrisonburg, VA 22807, USA; critchkl@jmu.edu; 4Division of Mental Health and Addiction, University of Oslo, Kirkeveien 166, 0450 Oslo, Norway; p.a.hoglend@medisin.uio.no (P.A.H.); randi.ulberg@medisin.uio.no (R.U.); 5Department of Psychiatry, Diakonhjemmet Hospital, Forskningsveien 7, 0370 Oslo, Norway

**Keywords:** SASB, FEST, transference work, object relations, process, outcome, psychodynamic, atmosphere

## Abstract

There is uncertainty concerning what the active ingredients in psychotherapy are. The First Experimental Study of Transference interpretations (FEST) was a randomized controlled trial of the effects of transference work (TW) in psychodynamic psychotherapy. Women with low quality of object relations (QOR) showed a large positive effect of transference work, while men with high QOR showed a slight negative effect. The present study aimed to expand the knowledge from the FEST by investigating the therapeutic atmosphere with Structural Analysis of Social Behavior (SASB). Two-way ANOVAs were conducted to investigate differences between SASB cluster scores between subgroups. The therapeutic atmosphere was characterized by Protect–Trust, Affirm–Disclose and Control–Submit. Multilevel modeling was used to assess the relationship between a therapist variable and outcomes for men and women. Contrary to expectations, no significant differences in therapeutic atmosphere between subgroups (with or without TW in women with low QOR and men with high QOR) were observed using the process measure SASB.

## 1. Introduction

What characterizes the in-session psychotherapeutic environment that promotes improved mental health? It is by now well established that psychotherapy is effective for many people who seek psychological treatment [1,2,3]. However, uncertainty still exists regarding the active, and most important ingredients, that make psychotherapy effective [4]. The elements that constitute psychotherapy can be divided into common factors and specific techniques. The term “common factors” refers to shared aspects of all effective treatments including alliance, empathy, expectations, cultural adaptation, and therapist differences [5]. Of these components, an alliance between therapist and patient is the most researched [3]. A specific technique in psychotherapy can be understood as a defined tool or method used by a therapist to promote effective therapy or positive change in the patient [6], in addition to the common factors.

### Theoretical Framework

Psychological difficulties often arise when interpersonal patterns are problematic and nonadaptive [7,8]. A patient’s inability to form stable and rewarding relationships is a major risk factor for the development and maintenance of psychopathology [9]. In mainstream clinical psychodynamic theory, it is maintained that the patients’ past relational history, affective experiences, and attachment patterns influence the ongoing interactions between patient and therapist [10,11]. How we relate to significant others is partly formed in early relationships with caregivers [8,12]. Psychological symptoms can be understood as embedded in cyclic, self-sustaining interpersonal patterns and transactions [13]. From the contemporary interpersonal perspective, neither the patients nor the therapist’s affective involvement in the therapeutic relationship is considered neutral [14].

Object relations theory emphasizes that internal representations of relationships with important individuals influence the way we shape relationships, select friends and partners, and understand and experience our social worlds. These internal representations are called internal object relations [10]. Transference can be defined as a tendency in which representational aspects of earlier important and formative relationships can be both consciously experienced and/or unconsciously ascribed to other relationships in the here and now [15]. Hence, transference may affect the here and now relationship, e.g., by distorted naiveté or skepticism and be of hindrance for real, trusting relationships both inside and outside the therapy room. Transference work is a specific technique where the therapist encourages the patient to explore their reactions to him/her. Working with the transference in the here and now may shed light on possible connections with the patient’s internalized learning history from previous relationships and thereby, facilitate the separation between what was from what is [16]. Quality of object relations (QOR) is another relevant term in this context. High QOR entails having a fundamental positive sense of self and relating to both positive and negative traits in others, while low QOR is characterized by dependency, exploitation, instability, and a lack of reciprocity and emotional investment. A person’s QOR can be measured using the Quality of Object Relations Scale (QORS) [17,18]. The QORS measures the patient’s life-long tendency to establish certain kinds of relationships with others, from mature to primitive.

The First Experimental Study of Transference interpretations (FEST) was a randomized controlled trial examining the effects of transference work (TW) in psychodynamic psychotherapy. The FEST had a strict dismantling design [16,19]. Unexpectedly, there was no main effect of transference work, but a significant interaction effect between transference work, quality of object relations (QOR), and gender, was uncovered [20]. Women with low QOR showed a large positive effect of transference work, while men with a high QOR showed a slight negative effect. On average, all subgroups improved.

The Structural Analysis of Social Behavior (SASB) [21] provides a lens to describe interpersonal and intrapsychic events and the coding system allows for an operationalization of these relations. SASB has proven useful in a variety of areas of interest [22]. Most relevant to this study is SASB’s application to psychotherapy process-outcome research. Henry et al. [23] found that greater levels of therapists Protecting and Affirming, and lower levels of Blaming were associated with high-change cases. Patient behavior of Disclosing was significantly more frequent in high change-cases, whereas Walling off and Trusting were significantly more frequent in the low-change cases. In a similar vein, Henry, Schacht, and Strupp [24] noted that patients in the poor outcome group were significantly more Watching and Managing (labeled Control in the present study) toward the therapist. They were also more Separating, Sulking, Walling off, and less Disclosing. A more recent study examined sessions from three variants of cognitive-behavioral therapy (CBT) [25]. They did not find interpersonal behaviors to be strong predictors of outcome in their sample. Von der Lippe et al.’s [26] analyses showed that stable hostile complementarity defined the negative change and nonchange therapies. Friendly complementarities predicted a positive outcome. Their results also indicated a negative effect of being “out of tune”, that is, lower correlations between therapist and patient communication. To summarize, previous studies using SASB have found communication characterized by high affiliation and high interdependence to be associated with positive psychotherapy outcomes and communication characterized by low affiliation and low interdependence to be associated with negative psychotherapy outcomes.

FEST found different outcomes for different patients, depending on their gender, QOR, and whether they received transference-based therapy or not. But what characterizes the therapeutic environment in these therapies? Examining the process between therapist and patients in sessions may shed light on the atmosphere in therapy which may contribute to the understanding of outcomes. Based on these results from FEST, we aim to investigate what characterizes the therapeutic atmosphere in good outcome psychodynamic therapy across time with and without TW in men with high QOR and women with low QOR as measured by SASB. Our hypothesis is that there is a difference in communication patterns between the subgroups. The differences in communication patterns found between the subgroups are thought to mediate change in the outcome measures across time.

## 2. Method

### 2.1. FEST

As methods and research design of the FEST previously have been described in detail [16,19], a summary of the methodology will follow.

#### 2.1.1. Patients and Therapists

The patients in the FEST sought psychotherapy for anxiety disorders, depressive disorders, personality disorders, and interpersonal problems. One hundred patients were included. Standard power calculation (endpoint analyses) indicated that moderate effect sizes (effect size = 0.55) could be detected for alpha levels of 0.05 with a power of 0.80. An alpha level of 0.10 was used for the moderator analyses and the subgroup analyses to balance the risk of type 2 errors [16]. Patients were assigned to seven highly experienced therapists who also served as clinical evaluators of other patients. A random assignment procedure was conducted and no differences between the groups were demonstrated on the pretreatment variables, including demographic, diagnostic, initial severity, personality, motivation, and expectancy [16].

#### 2.1.2. Treatment

The treatment consisted of one session weekly for one year. The sessions were audio-recorded. Treatment manuals of principles, not step-by-step procedures, were used for both treatment conditions [27]. In the pilot phase of the study, the therapists were trained for up to four years in order to enable them to provide treatment with a low to moderate level of transference work, and treatment without such interventions, with equal ease and mastery. For both treatment groups, psychotherapy was based on general psychodynamic treatment techniques, such as: focus on affects, exploration of warded-off material which often involved uncomfortable emotions, focus on current and past relationships, and interpretations of wishes, needs, and motives.

Half of the patients (*n* = 52) received dynamic psychotherapy with a moderate use of transference intervention (the transference work group). For the transference work group, the specific techniques presented in Table 1 were prescribed to the therapists [27]. In the comparison group (*n* = 48), these techniques were proscribed. Instead, the therapists consistently focused on interpersonal relationships outside of therapy as the basis for similar interventions, as opposed to the here and now relationship between patient and therapist.

Medication was used only for some patients with more severe depression or anxiety, and thus was associated with a slightly worse treatment outcome. Eleven patients in the transference group (21%) and nine patients in the comparison group (19%) were treated with antidepressant medication during therapy. The use of medication was well balanced across the two treatments and the subgroups and did not explain or change the results reported in the study [16].

#### 2.1.3. Assessment/Measures

Before randomization, all patients went through a psychodynamic interview [16,19]. The interviews were audio-recorded, and at least 3 evaluators did the scoring using the Psychodynamic Functioning Scale (PFS) [28,29] and the Quality of Object Relations Scale (QOR) [17,18]. The patients were also evaluated with self-report measures at the beginning, during, and at the end of treatment, and one and three years after termination of treatment. Interviews were not employed during treatment, but at pretreatment, the end of therapy, and at two follow-ups. All 100 patients were evaluated at the 3 year follow-up [16]. PFS [28,29] was the primary outcome measure in FEST. This clinician-rated measure consists of three relational subscales: quality of family relationships, quality of romantic and sexual relationships, quality of friendships; and three dynamic subscales: tolerance for affects, insight, and problem-solving capacity. The interrater reliability estimates of the PFS was 0.91, based on the average scores of four evaluations made by three raters [16,30]. QOR was the preselected moderator [17,18].

#### 2.1.4. FEST Results

The FEST did not show significant differences due to transference work. However, the patient-level quality of object relations (QOR) was found to be a significant moderator. That is, patients with low QOR benefited more from transference work than patients with high QOR [16]. This effect was stable three years after treatment termination [18] and seemed to be mediated by an increasing level of insight during treatment [31]. When patient gender was combined with the moderator QOR, a strong effect emerged: women with low QOR showed a large positive effect of transference work and men with high QOR showed a negative effect of transference work [32]. This interaction effect was found to be stable 3 years after treatment termination [33].

### 2.2. SASB

Structural Analysis of Social Behavior (SASB) was selected to code patient–therapist interactions. The model draws on developmental clinical knowledge and earlier interpersonal models as well as interpersonal theory, in order to describe interpersonal behavior. Relevant to this study is SASB’s application to psychotherapy process-outcome research, which is used as a lens to describe and operationalize interpersonal and intrapsychic events. The SASB methods are theory-neutral and have been used for assessing aspects of a variety of therapy approaches [34]. It envisions interpersonal and intrapsychic events on three surfaces that represent corresponding parenting and childlike behavior, as well as one representing expected self-concept as a result of the interaction between a child and their caregiver’s behavior [35].

In SASB, attentional focus is shown by separate surfaces. One has to consider if the focus is transitive and about you (e.g., therapist focuses on the patient) or if it is intransitive and about me (e.g., the patient focuses on him/herself). The different types of focus are described by two dimensions: Affiliation on the horizontal axis (friendly vs. hostile), and Interdependence on the vertical axis (sharing space vs. separating). Figure 1 shows a three-surface model. The third surface (italicized) is solely intrapsychic and not interpersonal and is therefore not included in the analyses or further discussions. The horizontal axis places maximum affiliation or sexuality on the right and maximum attack or murder on the left and the vertical places maximum separating on the top and maximum enmeshment at the bottom [36]. In other words, everything interactive is described in terms of underlying “primitive” basics of sexuality, aggression, dominance, and separate territory.

Two experienced therapists were trained to conduct SASB process coding (i.e., the here-and-now process between patient and therapist was coded). Three sessions from different phases of the therapy of each patient were selected for coding: the beginning of the therapy, mid-treatment, and late phase; 10 min × 3 of each of the three sessions, from the beginning, middle, and end of the session. The raters worked with the transcripts while listening to audio recordings for intonation and other audible nonverbal signals. The inter-rater reliability (weighted kappa) for the three assessments was on average 72 [32].

#### 2.2.1. Patients

Of the original sample of 100 patients, 42 were selected to have 3 therapy sessions coded using the SASB instrument. The patients selected were from two subgroups; men with high QOR (N = 21) and women with low QOR (N = 21). This means that the two characteristics are completely confounded, and the contrast involves the maximum effect size in the gender-moderated effect of transference work on outcome. With a few exceptions, most pretreatment patient characteristics were evenly divided in the TW group and the non-TW group for both genders (see Table 2).

#### 2.2.2. Statistical Analysis

A series of mixed ANOVAs were conducted in order to select a subset of SASB dimensions that could subsequently be used as predictor variables in multilevel models. We aimed to investigate whether different levels of SASB domains across the two groups could account for the different PFS outcome scores observed in the FEST. Time and treatment type were therefore selected before conducting two-way mixed ANOVAs, including one factor (time) within patients, and another (treatment) between patients. The three ratings within each session were averaged into one score as the variation between, rather than within sessions; this was of main interest. A mean from the three measuring points was created and transformed into a standardized score. The transitive focus where the patient focuses on the therapist and the intransitive focus where the therapist focuses on him/herself (i.e., unexpected reversal of roles for a therapeutic encounter), was scarcely coded and was excluded from the analysis. After selecting the potential SASB domains that differed between subgroups, longitudinal analyses of PFS were conducted.

Multilevel modeling was used to assess change over time and possible interaction effects and a series of models assessing the impact of the type of treatment and patient–therapist communication were fitted. We started with an empty model, before testing more complex models. In total, we fitted five models with an increasing number of fixed effects, such as the type of treatment (therapy with/without transference work), time, and therapist Control, in addition to the interaction between these predictors. Therapist Control was added to investigate therapist Control effects on PFS-scores across time in the two subgroups respectively. Due to the complete confounding between patient gender and the quality of object relations, the data for males (N = 21) and females (N = 21) were analyzed separately. An alpha level of 0.10 was chosen a priori for all subgroup analyses in the FEST [16] and was also chosen in the present study. All analyses were conducted using SPSS (version 25) [40].

## 3. Results

### 3.1. Line Charts

Line charts were made using SASB mean interaction scorings based on transcripts and audio recordings 3 × 10 min. These include both genders in both groups, rather than only the subsets used in multilevel models. In these visual representations, three types of patient–therapist interaction stood out; Protect–Trust, Affirm–Disclose, and Control–Submit. All communications between therapist and patient were predominantly classified into these three categories, both with women and with men. Virtually none of the therapist or patient statements were coded as one of the remaining five clusters of behavior. Plots of SASB coding in men and women in both the TW group and the non-TW group indicated little systematic variability over time in scores across three sessions from the beginning, middle, and the end of therapy (see Figure 1). However, therapist Control seemed to be slightly higher in women and men in the TW group compared to women and men in the non-TW group.

Therapist scores were selected for further exploration as the complementarity between therapist and patient SASB cluster scores was high and as the dataset included a randomized variable (transference work), which is a technique implemented by the therapist. Line charts with therapist SASB cluster sores with women and with men illustrate the almost identical therapist behavior between these two groups (see Figure 2).

### 3.2. ANOVA

We conducted mixed ANOVAs with the factors *time* and *treatment group*. First, as a check for significant interpersonal differences for gender between each arm of the dismantling study, we compared women in the TW group and women in the non-TW group. The same was done for the men. Women in the TW group were compared to men in the non-TW group and vice versa. Both patient and therapist variables were analyzed. The results indicated no significant differences in the amount or change over time of therapist or patient communications in all combinations of groups (*p* > 0.1). When analyzing the TW group (including both women and men) vs. the non-TW group (including both women and men), a difference was found only with respect to the Control–Submit interaction, with *p* < 0.1. Based on this result, therapist Control codes were selected for further analyses.

### 3.3. Multilevel Modeling

The results from multilevel modeling (MLM) indicated the same trend as the results from the main analyses in the FEST: a positive relationship between transference work and outcome for women with low QOR and a negative one for men with high QOR. No significant effect of therapist Control on PFS outcome scores were found for either subgroup.

## 4. Discussion

The results from the present study indicate that the therapeutic atmosphere was characterized by Protect–Trust, Affirm–Disclose, and Control–Submit. The overall positive SASB coding (high affiliative control and affiliative autonomy) is consistent with previous research connecting these interactions to a good outcome [23,24]. Based on the SASB framework, therapist interpersonal style, such as communication characterized by Protect and Affirm, evokes complementary behavior in the patient, leading to a therapeutic atmosphere identified by high affiliation. Previous studies using SASB have found a therapeutic atmosphere characterized by high affiliation and high interdependence to be associated with positive psychotherapy outcomes. It is reasonable to assume that this affiliative environment discovered in the present process study can explain the positive psychotherapy outcome for all patients in the study.

In an article by Constantino [22], therapeutic alliance is understood in terms of interpersonal processes, and SASB’s wide-ranging applications is presented to observe the therapist–patient relationship, therapy process, and outcome. Positive complementarity in SASB coding is associated with positive outcomes [23,26]. As there is a high degree of complementary between patient and therapist on Protect–Trust and Affirm–Disclose in the present study in addition to a low degree of hostility, one might consider the SASB profile discovered in the current study as an indication of a good alliance, replicated for each subgroup. Complementarity interactions also tend to sustain relationships and are assumed to enhance security and reinforce current interaction patterns [41]. In addition, studies indicate that therapists’ non-defensive responses to client negativity or hostility are critical to maintaining a good alliance [42]. In the FEST, Høglend et al. [43] found that for patients with low QOR and more severe psychopathological struggles, the effect of transference work was positive independent of the quality of the alliance. The importance of both alliance and TW is an indication of the growing consensus in clinical psychology that both common and specific factors shape therapeutic outcomes [11,44,45]. In interpersonal terms, this suggests that the content discussed in psychotherapy is important in ways that are separable from the in-session relational process or manner of discussion. The literature suggests that both are important.

Small differences were found between the subgroups concerning the most frequently coded SASB cluster scores. Affirm–Disclose and Protect–Trust were almost identical within the four groups, while a difference in Control–Submit was apparent between the TW group and non-TW groups. This was surprising for two main reasons. Firstly, the frequency of Control–Submit coding was less substantial compared to the affiliative interaction patterns Affirm–Disclose and Protect–Trust. However, it is clear that a Control–Submit pattern played a part in the therapeutic dialogue, presumably mostly in the TW group. It could be expected that TW, where the therapist to a higher degree takes the lead in communication when commenting on transference, would color the interpersonal interaction and therapeutic atmosphere to a higher degree than therapy without TW. The presence of therapist Control did not seem to have an impact on the outcome. The small effect could partly be explained by the small number of patients in each subgroup. Secondly, we expected to see more variance in the SASB coding in the groups that would reflect the different subgroup characteristics and outcomes after treatment. Based on this, it was considered plausible that the amount of positive and negative interactions between the group with a positive outcome (women with low QOR in the TW group) and a less positive outcome (men with high QOR in the TW group) would be visible in the SASB coding.

Several studies show an interaction between interpersonal problems and personality disorders [46]. As the sample of the present study is made by comparing two groups with differing degrees of interpersonal problems/personality pathology (i.e., indexed by QOR), one could expect to observe different interpersonal processes in the group, which is characterized by a higher degree of interpersonal problems, compared to a group where these problems are present to a lower degree. We would also expect to observe a difference regarding transference work and no transference work. On the other hand, it is relevant to mention that this study is based on average scores for groups, and thus has a nomothetic perspective. Therefore, individual variation is disguised behind average cluster scores.

The results suggest that there must be something SASB, as used in the present study, failed to capture. Our understanding of the process and different outcomes in this population was not illuminated by the present analyses. Consequently, the reason why women with low QOR in this study benefitted more from psychotherapy with transference work than psychotherapy without transference work is still not clear. One possibility is that the interpersonal content and accuracy of transference interpretations was not assessed. For example, if a therapist interprets a patient as feeling neglected by him/her, when in fact they are feeling criticized, this has been linked in prior research to reduced outcomes [47,48]. It is possible that the impact of TW on relational processes is itself a function of how acceptable/accurate the interpretations are from the point of view of patients. It is also possible that the therapeutic relationship is only part of the picture for determining patient outcomes, which in this case appears to have occurred for all groups in the context of a near-optimal therapeutic process (i.e., few if any hostile process codes predictive of negative outcomes in prior work by Henry et al. [23,24] and Von der Lippe et al. [26])

It is reasonable to believe that more variation would be revealed if a more complex model had been used. According to Benjamin [34], the full model is best for the analysis of complex situations that require high precision. Moreover, studies by Henry et al. [23,24] found more variation in SASB clusters using the simplified models than what was found in the present study. In addition, other studies using SASB [23,26,49] included data on the momentary response to patients’ and therapists’ statements. The present study did not include this type of data. This means that the study of complementarity was limited to an analysis on mean scores, as information on speaking turns was not available. More specifically, it was not possible to determine whether patient trusting, disclosing, and submitting elicited the complementary response in the therapist, or whether the interaction originated from the therapist’s communication. With a larger population and more statistical strength, stronger (or different) relationships might have been observed. The study of differences between groups instead of between patients alone might have led to missing variance. Another potential source could be the fact that only three sessions (3x10 min per session) were coded with SASB; understandably so, as it is a time consuming and challenging task. However, more variability would likely be visible if more sessions were coded. Both groups in the FEST had a positive outcome, and therefore the differences might not be as apparent as they would if the study included a good outcome and bad outcome group, as found in earlier studies using SASB [23,24,26]. Similar to the present study, a more recent study on CBT therapies did not find interpersonal behaviors measured by SASB to be strong predictors of outcome [25].

### 4.1. Limitations

The analyses and results and are based on data from a limited number of participants; a selection of 42 patients originally form 100 patients from the FEST. Additionally, the selection of participants from the original study only consists of women with low QOR and men with high QOR. This limits the generalizability, as the sample does not include men with low QOR and women with high QOR. The complete confounding of the variables of QOR and patient gender also leads to the impossibility of concluding on the basis of gender or QOR separately. Pretreatment characteristics was nearly, but not completely, evenly divided in women in the TW group compared to women in the non-TW group, and for men in the TW group compared to men in the non-TW group. This means that there is a possibility that the interpretations of results could have been affected by different pre-treatment characteristics in the subgroups. Possibly important data on work environment, physical exercise, and somatic health was not collected and subsequently not considered in the analyses, hence we do not know if these variables were successfully randomized. In addition, the therapists in FEST are very experienced, had worked together for a long time, and were part of the research team. Our findings cannot be generalized to novice therapists or to more intensive or longer-term therapy.

### 4.2. Future Research

Interpersonal processes are key variables to study in order to reach a better understanding of the active and most important ingredients that make psychotherapy effective. In the present study, a small difference in SASB cluster scores was found only concerning the Control–Submit interaction in the TW vs. non-TW groups. No significant effect of therapist Control on outcome was found for either subgroup. However, the analyses and results were based on data from a limited number of participants. It is therefore recommended that future research on process and outcome mechanisms in psychotherapy is built upon a larger number of participants. In future research on therapeutic atmosphere and the interpersonal elements of TW, therapist Control might be an interesting element to investigate further. The reliability of less experienced therapists could be investigated in future research. Future research could also explore possible discrepancies between therapists’ self-reported feelings and interactions with the patients, as rated by external observers and whether this effects the therapeutic atmosphere and outcome.

## 5. Conclusions

The present study aimed to investigate what characterizes the therapeutic environment in psychodynamic therapy using transference work. The analyses were based on therapy processes from the FEST using a simplified version of SASB. The results indicate that the therapeutic atmosphere was characterized by Protect–Trust, Affirm–Disclose and Control–Submit. This is partially in accordance with previous findings on therapy processes using the SASB model. We have sought to fill part of the knowledge gap concerning why a specific technique works for some but not others in psychotherapy. Contrary to expectations, no significant differences in SASB cluster scores between subgroups (high QOR men and low QOR women offered therapy with or without TW) were observed. The study did not, as expected, enhance our understanding on why transference work was of such importance for women with low QOR. The present study contributes to the research field by investigating and clarifying what the active ingredients of psychodynamic therapy are, and what characterizes the therapeutic atmosphere and interaction in the relevant subgroups and in transference-based therapy.

## Figures and Tables

**Figure 1 ijerph-17-04105-f001:**
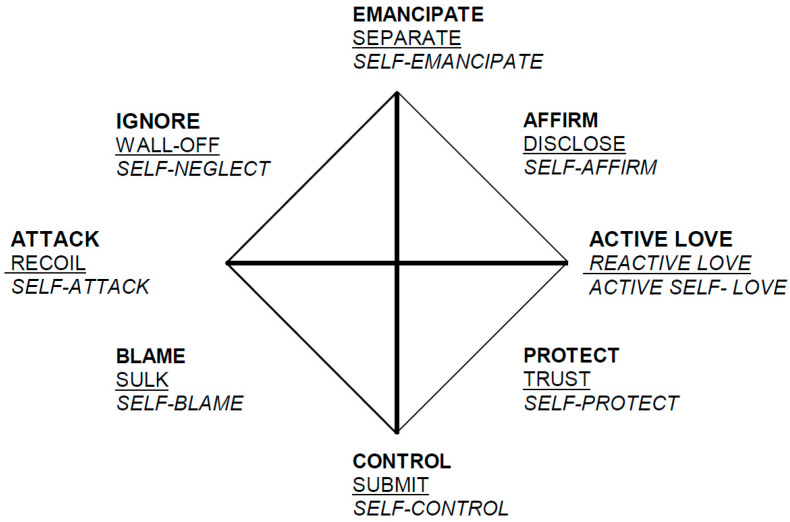
Simplified cluster version of the Structural Analysis of Social Behavior (SASB-model). From Benjamin [37] Guilford Press, reproduced by permission. Copyright Guilford Press.

**Figure 2 ijerph-17-04105-f002:**
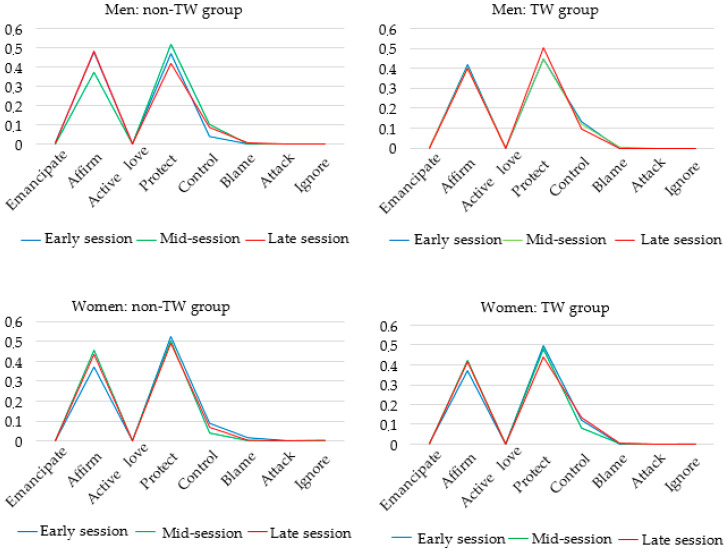
The four plots represent therapist SASB cluster scores in each subgroup. Blue line: session early in the treatment, green line: mid-treatment, red line: late phase of treatment. SASB clusters along the x-axis and the SASB the proportion of cluster scores (ranging from 0–1) on the y-axis.

**Table 1 ijerph-17-04105-t001:** The five categories of transference work (TW) defined in the First Experimental Study of Transference interpretations (FEST).

1. The therapist was to address transactions in the patient–therapist relationship.
2. The therapist was to encourage the exploration of thoughts and feelings about the therapy and therapist.
3. The therapist was to encourage patients to discuss how they believed the therapist might feel or think about them.
4. The therapist was to include him/herself explicitly in interpretive linking of dynamic elements (conflicts), direct manifestations of transference, and allusions to the transference.
5. The therapist was to interpret repetitive interpersonal patterns (including genetic interpretations) and link these patterns to transactions between the patient and the therapist.

Based on Høglend [27].

**Table 2 ijerph-17-04105-t002:** Pre-treatment characteristics for patients receiving dynamic psychotherapy for 1 year with and without transference work. Characteristics for women/low quality of object relations (QOR) and men/high QOR are shown separately.

	Low QOR Women (N = 21)				High QOR Men (N = 21)			
Characteristic	TW Group (N = 11)		Non-TW Group (N = 10)		TW Group (N = 11)		Non-TW Group (N = 10)	
	Mean	SD	Mean	SD	Mean	SD	Mean	SD
Age	32.6	7.3	34.7	9.5	40.7	9.8	40.4	9.8
IIP-C *	1.2	0.3	1.3	0.5	1.2	0.6	1.2	0.5
GSI (SL-90) **	1.2	0.8	1.2	0.4	1.1	0.6	0.8	0.5
PFS ***	61.8	1.8	59	4.8	65.1	3.7	67.6	3.1
PD-Criteria (SCID-II)	11.5	6.3	11.5	7.6	7.3	6.4	3.9	3.1
	**N**	%	**N**	%	**N**	%	**N**	%
Single	6	54.5	4	40	1	9	1	10
Caucasian	11	100	10	100	11	100	10	100
Personality Disorder	7	63.6	6	60	3	27.3	1	10
Active employment	5	45.5	7	70	10	90.9	6	60

* Inventory of Interpersonal Problems [38], ** Global Severity Index [39], *** Psychodynamic Functioning Scale [28,29].

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
