# Peer review of "Therapeutic Atmosphere in Psychotherapy Sessions"

_ijerph, 2020, doi:10.3390/ijerph17114105_

Round 1

Reviewer 1 Report

In this manuscript, the authors present therapeutic atmosphere in psychotherapy sessions, these are some of my comments

It is an interesting paper with few novel results.

Pease correct the phrase”…Common factors are those factors that all therapies are considered to have in common…”, it is absurd.

Why one hundred of patients?

Were patients with similar disorders included in the two groups?

Was any type of pharmacological treatment included?

Was exercise taken in account?

Were the patients employed or unemployed, was their work environment taken into account?,

Author Response

Dear reviewer, 

Thank you so much for the helpful comments and input. In our manuscript we have highlighted the changes that have been made according to the comments.

Please find our response to the comments below. 

  • Pease correct the phrase”…Common factors are those factors that all therapies are considered to have in common…”, it is absurd. 
    • We have rephrased this as “The term common factors refer to shared aspects of all effective treatments including alliance, empathy, expectations, cultural adaptation, and therapist differences”. Please see page 1, lines number 41-43.
  • Why one hundred of patients?
    • We have added information on the standard power calculation (endpoint analyses).Please see page 3, lines 115-118
  • Were patients with similar disorders included in the two groups?
    • A random assignment procedure was conducted and no differences between the groups were demonstrates on the pretreatment variables, including demographic, diagnostic, initial severity, personality, motivation, and expectancy. We have added this information to section 2.1.1. Please see page3, lines 120-122. 
  • Was any type of pharmacological treatment included?
    • Eleven patients in the transference group (21%) and nine patients in the comparison group (19%) were treated with antidepressant medication during therapy. This information has now been added to section 2.1.2 of the manuscript. Please see page 3, lines 139-143 
  • Was exercise taken in account?
    • The randomization was successful on other measures, so hopefully also concerning exercise. However, exercisewas not considered in the FEST study or the present study. 
  • Were the patients employed or unemployed, was their work environment taken into account?
    • We have added information on the patients employment status in the manuscript and table 2. Please sepage 6, line 213 for this information. Work environment was not taken into account 

Reviewer 2 Report

Dear authors,

Thank you so much for your submission and valuable work on psychotherapy. Please find my comments below to improve your manuscript.

1. Please carefully revise the format of the whole manuscript, including main text, tables, figures and references, in the next version according to the requirement of this journal. I think it is necessary before your final submission.
2. I would like to see one detailed section named as the literature review or theoretical framework in your manuscript. You can describe why you want to conduct this study with the development of your theoretical hypothesis and further verification if you really conducted a psychological study.
3. Please describe your research question more specifically. Try to keep your research question short, concise and arguable.
4. Even though the information about the FEST has been stated in other publications, you have to restate it here as far as you can.
5. The representativeness of your sampling is required to report in this manuscript. Otherwise, your readers cannot judge and understand how important and how valuable your study was.
6. What do you mean by this sentence "In the present study a two-surface model of SASB was used in SASB, attentional focus is
156 shown by separate surfaces. "?
7. English proofreading is extremely needed.
8. Please add detailed information about SASB in your manuscript. Please let your readers know about why this method is appropriate for your study.
9. How did you conduct your Multilevel modelling? Any information about your results after your analysis? Have you ever presented your findings in your manuscript?!

Author Response

Dear reviewer, 

Thank you so much for the helpful comments and input. In our manuscript we have highlighted the changes that have been made according to the comments. 

Please find our response to the comments below.

  • Please carefully revise the format of the whole manuscript, including main text, tables, figures and references, in the next version according to the requirement of this journal. I think it is necessary before your final submission.
    • We have reviewed the manuscript according to the journal requirements and made necessary changes.
  • I would like to see one detailed section named as the literature review or theoretical framework in your manuscript. You can describe why you want to conduct this study with the development of your theoretical hypothesis and further verification if you really conducted a psychological study.
    • We have now clarified with a section(1.1.) titled theoretical framework on page 2 line 47. In this section we have presented interpersonal difficulties as a common struggle for patients with psychological disorders. Further, we introduce psychodynamic theories concerning object relations and transference, that are relevant aspects of patients’ social life and relation to the therapist. Descriptions of FEST and SASB follow to illustrate how interpersonal aspects of the therapy relation and atmosphere can be investigated in process-outcome studies. Our aim is to use SASB to further explore the interaction and atmosphere between the therapist and patients from the FEST using SASB to investigate process-outcome in psychodynamic therapy.  
  • Please describe your research question more specifically. Try to keep your research question short, concise and arguable.
    • We have reformulated our research questions according to your comment and attempted to make it more concise and arguable. Please see page 3, lines104-108.  
  • Even though the information about the FEST has been stated in other publications, you have torestate it here as far as you can.  
    • We have added information about the power analyses, the patients and the random assignmentfrom FEST. Please see section 2.1.1, page 3 line 115-118 line 120-122.  
  • The representativeness of your sampling is required to report in this manuscript. Otherwise, your readers cannot judge and understand how important and how valuable your study was.
    • We have addedmore information on pretreatment patient characteristics in table 2 and information about the randomization: A random assignment procedure was conducted and no differences between the groups were demonstrates on the pretreatment variables, including demographic, diagnostic, initial severity, personality, motivation, and expectancy. Please see section 2.1.1 page 3, line 120-122.
  • What do you mean by this sentence "In the present study a two-surface model of SASB was used in SASB, attentional focus isshown by separate surfaces. " 
    • This sentence has been reformulated to better explain our point about the separate surfaces. Please see section 2.2, page 5, line 180. 
  • English proofreading is extremely needed.
    • Thorough proofreading is conducted.  
  • Please add detailed information about SASB in your manu Please let your readers know about why this method is appropriate for your study.
    • More detailed information has been added about SASBin section 2.2 on page 4-5 lines 171-179 and 186-189 to make it more clear how SASB is relevant to the present study.  
  • How did you conduct your Multilevel modelling? Any information about your results after your analysis? Have you ever presented your findings in your manuscript?!
    • The multilevel modeling was conducted as following: 
      The following steps were identical for men with high QOR (N=21) and women with low QOR (N=21) (see tables 2 and 3); We started with an empty model, model 1, before testing more complex models. We ran analyzes with and without a random effect of time. A model without a random effect of time fitted the data better according to AIC-criterion and only a fixed effect of time was therefore included in model 2. The absence of random effects of time indicated that the participants PFS scores changed in a similar way across time. Two fixed effects were added in model 3: 1) treatment (therapy with/without transference work), 2) the interaction between type of treatment and time. The results indicated support for a model including these predictors. We wanted to see if a model including the therapist Control would explain even more of the variance and if the effect of therapist Control on outcome across time was negative or positive for women and for men. We included the fixed effects; Control and the interaction between Control and time to model 4. Our hypothesis was that there would be a positive effect of Control on outcome women and a negative effect of Control on outcome for men. As a final step, we included a three-way interaction between time, Control and type of treatment to model 5. The question of interest was how the interaction between time and treatment group (time x treatment) would be affected by this inclusion, as a decreasing score would imply that the effect of therapist “Control” on outcome would be different in the TW-group compared to non-TW group, in the group of women and men separately." 

      We have revised accordingly and added a shortened version of this to the manuscript (please see section 2.2 page 6, lines 228-233. As the findings from the multilevel models were insignificant, we have prioritized to devote less room in the manuscript to this topic.  The findings have never been presented previously neither in publications nor in any presentation. 

Round 2

Reviewer 1 Report

Although the manuscript is better, it still has important weaknesses such as not including the exercise and work environment among others, the authors should emphasize these weaknesses in their conclusions.

Author Response

Dear Reviewer

Thank you so much for the helpful comments in this second round of revisions. In our manuscript we have highlighted the changes that have been made according to the comments. Please find our response to the comments below.  

Although the manuscript is better, it still has important weaknesses such as not including the exercise and work environment among others, the authors should emphasize these weaknesses in their conclusions. 

  • We have added a section called Limitations. Please see page 9-10 line 367-381In the FEST, the study on which the present study is based, your comments are highly relevant, unfortunately these variables were not included in the FEST-study. From listening to sessions, we know that work environment was discussed during several of the therapies, however this material is not available for further analyses. 

Reviewer 2 Report

  1. Your abstract is not well structured to present your research.
  2. Please provide the high quality of figures in your manuscript.
  3. What are the contributions of your work to this research field?
  4. What are the research gaps that you have closed?
  5. What are the criteria for your statistical analysis?
  6. How did you control the bias and confounders in this study?
  7. How would you make use of your findings in policies-making, research or practices in future?

Author Response

Dear Reviewer

Thank you so much for the helpful comments in this second round of revisions. In our manuscript we have highlighted the changes that have been made according to the comments. Please find our response to the comments below.  

Your abstract is not well structured to present your research.  

  • Thank you for pointing this out. We have tried to improve accordingly 

Please provide the high quality of figures in your manuscript 

  • Figure 1 has been replaced with a figure of higher quality, please see page 4. 

What are the contributions of your work to this research field?  

  • The present study shows that there may be only minor differences in therapeutic atmosphere in very different patient groupswith therapists using different techniques, leading to different outcomes for the patients 

What are the research gaps that you have closed? 

  • We have sought to fill part of the knowledge gap concerning why a specific technique works for some, but not for others in psychotherapy. As the minor differences between therapeutic atmosphere int the different subgroups did not seem to mediate differences across time, the study did not, as expected, enhance our understanding of why transference work was of such importance for women with low QOR.  

How would you make use of your findings in policies-making, research or practices in future? 

  • We have attempted to clarify this in the section 4.3 Future research and section 5. ConclusionPlease see pages 9-10.  

What are the criteria for your statistical analysis?  

  • Multilevel modelling was chosen for the statistical analysis because of its suitability on psychotherapy research with longitudinal data. This because of the model's flexibility in dealing with time and the possibility of including both time-specific and time-invariant predictors, and their interactions. MLM does not require homogeneity of variance and it takes the sampling hierarchy into account, and it can analyze incomplete data (Field, 2013; Quené & van den Bergh, 2004). 

How did you control the bias and confounders in this study? 

  • This is important, and the first control was that the randomization was deemed successful following a strict randomized controlled study paradigm. All therapists treated patients from both groups. No therapist ratings of their own patients were included in any of the statistical analyses. Also, all 100 patients came to the three years follow-up. The therapists may have been biased in favor of transference work and felt that the patients deprived of this technique were getting less than optimal treatment. This possibility cannot be ruled out with absolute confidence. On the other hand, when polled at treatment termination, therapists were clearly in favor of using relationship work with 65% of the patients with mature object relations, but only with 50% of the patients with more severe personality pathology (Høglend, Hersoug, et al., 2011). In coding SASB process codes, two therapists were trained to conduct SASB process coding. The inter-rater reliability (weighted kappa) for the three assessments was on average .72.  Hence this is a study with rather high internal validity compared to most psychotherapy studies. That the therapists in FEST are very experienced and had worked together for a long time, in addition to be part of the research team will influence the external validity, where therapists usually are a more heterogenous group. Hence, the therapeutic atmosphere will probably vary more.   

Round 3

Reviewer 1 Report

No comments